# PURPL and NEAT1 Long Non-Coding RNAs Are Modulated in Vascular Smooth Muscle Cell Replicative Senescence

**DOI:** 10.3390/biomedicines11123228

**Published:** 2023-12-06

**Authors:** Clara Rossi, Marco Venturin, Jakub Gubala, Angelisa Frasca, Alberto Corsini, Cristina Battaglia, Stefano Bellosta

**Affiliations:** 1Department of Pharmacological and Biomolecular Sciences “Rodolfo Paoletti”, Università degli Studi di Milano, 20122 Milan, Italy; clara.rossi@unimi.it (C.R.); jakub.gubala@unige.ch (J.G.); alberto.corsini@unimi.it (A.C.); 2Department of Medical Biotechnologies and Translational Medicine (BIOMETRA), Università degli Studi di Milano, 20122 Milan, Italy; marco.venturin@unimi.it (M.V.); angelisa.frasca@unimi.it (A.F.); cristina.battaglia@unimi.it (C.B.)

**Keywords:** aging, biomarkers, lncRNA, NEAT1, PURPL, RRAD, senescence, smooth muscle cells

## Abstract

Cellular senescence is characterized by proliferation and migration exhaustion, senescence-associated secretory phenotype (SASP), and oxidative stress. Senescent vascular smooth muscle cells (VSMCs) contribute to cardiovascular diseases and atherosclerotic plaque instability. Since there are no unanimously agreed senescence markers in human VSMCs, to improve our knowledge, we looked for new possible senescence markers. To this end, we first established and characterized a model of replicative senescence (RS) in human aortic VSMCs. Old cells displayed several established senescence-associated markers. They stained positive for the senescence-associated β-galactosidase, showed a deranged proliferation rate, a dramatically reduced expression of PCNA, an altered migratory activity, increased levels of TP53 and cell-cycle inhibitors p21/p16, and accumulated in the G1 phase. Old cells showed an altered cellular and nuclear morphology, downregulation of the expression of LMNB1 and HMGB1, and increased expression of SASP molecules (IL1β, IL6, IL8, and MMP3). In these senescent VSMCs, among a set of 12 manually selected long non-coding RNAs (lncRNAs), we detected significant upregulation of PURPL and NEAT1. We observed also, for the first time, increased levels of *RRAD* mRNA. The detection of modulated levels of RRAD, PURPL, and NEAT1 during VSMC senescence could be helpful for future studies on potential anti-aging factors.

## 1. Introduction

The proportion of the world’s population aged 60 years or more is expected to double in the next four decades, and the WHO estimates that one in six people, or 2.1 billion, will be over age 60 by 2030 [1]. Aging plays a previously unrecognized, significant role in the pathophysiology of the human body [2]: it is associated with progressive degeneration of tissues, negatively impacting the structure and function of vital organs, and it is among the most important known risk factors for chronic diseases (such as atherosclerosis or neurodegeneration). Aging includes several diverse mechanisms such as oxidative stress, genomic instability, progenitor cell exhaustion or dysfunction, telomeric and epigenetic changes, altered nutrient sensing, mitochondrial dysfunction, chronic low-grade inflammation, altered protein homeostasis, fibrosis, microbiome dysregulation, and cellular senescence [3]. Senescence of various cell types, including endothelial cells, vascular smooth muscle cells (VSMCs), macrophages, and T cells, has been implicated in the pathogenesis of degenerative diseases such as atherosclerosis. VSMCs, as basic components of the vascular wall and the sole cell types in the arterial medial layer, play critical roles in vascular physiological functions [4,5], are the major source of atherosclerotic plaque cells, and contribute to plaque development and progression [6].

At the cellular level, senescent cells may form throughout an entire human lifespan, accumulating and secreting different factors that may cause deleterious effects on surrounding tissues [1]. Cellular senescence is a multi-step process first described by Hayflick as causing cells to no longer divide, having exhausted their replicative potential [7,8]. Senescence is characterized by cell-cycle arrest in the G1 or G2 phase [9], resulting from the silencing of proliferation-promoting genes upon activation of the tumor suppressor gene p53 and the cyclin-dependent kinase inhibitor (cdki) p21, and it is then stabilized by activation of the cdki p16 and hypophosphorylation of the retinoblastoma protein [8]. Cellular senescence can also be induced by various stimuli such as oncogene activation (oncogene-induced senescence) or DNA damaging agents and oxidative stress, called ‘stress-induced premature senescence’ [10].

Besides permanent growth arrest, senescent cells display morphological and functional changes, including flattened and hypertrophic morphology, nuclear enlargement, and expression of the ‘senescence-associated β-galactosidase’ (SA-β-gal), a pH-sensitive enzyme whose activity reflects an increased lysosomal mass [5]. Senescent cells stimulate reactive oxygen species (ROS) production, thus inhibiting cell proliferation [11]. Moreover, they also show a ‘senescence-associated secretory phenotype’ (SASP), with the production of proapoptotic and pro-fibrotic factors [12], and the secretion of pro-inflammatory cytokines (e.g., IL6 and CXCL8 (IL8)), chemokines (e.g., CCL2), and proteases (e.g., matrix metalloproteases (MMPs)), thus contributing to tissue inflammation [5,13].

Cellular senescence program initiation and sustainment are based on transcriptional and post-transcriptional, as well as epigenetic, changes, involving not only proteins but also different types of RNAs, including long non-coding RNAs (lncRNAs) [14,15,16,17]. LncRNAs are a wide and heterogeneous class of RNA molecules longer than 200 nucleotides with a fundamental role in the control of gene expression through different mechanisms [16,17] and are known to be involved in key biological processes [18,19]. More specifically concerning aging, several lncRNAs are involved in different hallmarks of senescence, among which are cell-cycle arrest, apoptosis, telomere stability, and inflammation [20]. Despite mounting evidence linking lncRNAs to senescence, only a few of them have been associated with the formation of senescent VSMCs so far [21,22].

To further our understanding of senescence in VSMCs, we evaluated the expression of manually selected senescence-associated genes and lncRNAs in a cellular model of replicative senescence in human VSMCs that we first carefully characterized by employing multi-biomarker approaches. Our investigation led to the discovery of an enhanced expression of PURPL, NEAT1, and RRAD in our cellular model.

## 2. Materials and Methods

### 2.1. Cell Cultures

Human aortic vascular smooth muscle cells (VSMCs) (PCS-100-012, ATCC, Manassas, VA, USA, received at 2nd passage) were cultured in ATCC Vascular Cell Basal Medium (PCS-100-030, ATCC; 500 mL added with 500 µL ascorbic acid, 500 µL rh EGF, 500 µL rh insulin and rh FGF-b, 25 mL glutamine), 5% fetal bovine serum (FBS, ATCC Vascular Smooth Muscle Growth Kit), and 5 mL Penicillin-Streptomycin 100× (Euroclone, Milan, Italy). After reaching 80–90% confluency, cells were detached with trypsin, diluted 1:2 or 1:3, and reseeded in fresh medium. In our experiments, VSMCs were used after 5–7 passages as young (proliferating cells), and after 15–17 passages as old (non-proliferating cells) to represent different stages of replicative senescence. Cultures were maintained at 37 °C in a 5% CO_2_ incubator. After the 15th–17th passage, cells reached the proliferative arrest and did not attach to cell culture plates anymore and died. Therefore, the 15th–17th passage was the limit we could not overcome in passing the VSMCs.

In some preliminary experiments, we used VSMCs after 9–11 passages to evaluate an intermediate senescence state. These cells were still proliferating and, therefore, we decided not to include the data in the present manuscript.

### 2.2. Senescence-Associated β-Galactosidase Staining

Young and old VSMCs were plated in 24-well plates at a density of 2 × 10^4^ cells/well. After 3 days, the activity of senescence-associated-β-galactosidase (SA-β-gal) was evaluated by staining VSMCs with the Senescence Cells Histochemical Staining Kit (CS0030, Sigma-Aldrich, St. Louis, MO, USA), following the manufacturer’s instructions [13].

### 2.3. Cell Proliferation

VSMCs were seeded in 24-well plates at a density of 3 × 10^4^ cells/well. After 24 h, cells were incubated with a medium containing 0.4% FBS to be synchronized at the G0 phase of the cell cycle. After 72 h, control dishes were counted with a Coulter Counter (Beckman Coulter, Life Scientific, Milan, Italy) and this was considered the “basal” number of cells at T0. Subsequently, the medium was removed and replaced with 10% of FBS. At different time points (up to 96 h), cell number was measured and compared to the T0. Results were also used to calculate the doubling time [23].

### 2.4. Cell Migration

VSMCs were seeded in 24-well plates and, after reaching 80–90% confluence, were detached and 2 × 10^4^ cells were plated in the upper chamber, which has a membrane with 8 μm pores separating it from the lower chamber where PDGF 5 ng/mL in 0.4% serum medium was added as chemoattractant. After 6 h, migrated cells were stained with a Diff-Quik staining set, as previously described [24]. The number of migrated cells was counted in five randomly chosen areas of the membranes at 10× magnification.

### 2.5. Cell Cycle Measurement

The cell cycle was determined after seeding VSMCs in 12-well plates and reaching 80–90% confluence. Then cells were washed with PBS, trypsinized, and fixed with ice-cold ethanol 66%. After fixation, cells were subsequently collected and stained with a propidium iodide flow cytometry kit (Abcam, ab139418, Cambridge, UK). The cell-cycle phases were analyzed with a NovoCyte 3000 Flow Cytometer (Agilent, Santa Clara, CA, USA) using the NovoExpress software v.1.3.3.

### 2.6. Immunofluorescence Analysis and Nuclear/Cell Size Measurement

Young and senescent VSMCs were seeded on glass coverslips in a 24-well plate. After 3 days in culture, cells were fixed with 4% paraformaldehyde in 10 mM PBS for 30 min at room temperature and then washed three times with 10 mM PBS. Cells were permeabilized at 4 °C in PBS containing 0.1% triton X-100 for 3 min and blocked with 5% BSA in PBS for 15 min. Then, cells were incubated with the primary antibody anti-RRAD (Thermo-Fisher, Waltham, MA, USA) in 0.2% BSA in PBS, overnight (4 °C) and then were incubated with Alexa Fluor 488-conjugated secondary antibody (Invitrogen, Waltham, MA, USA) for 1 h and washed with PBS. To evaluate nuclear and cell size changes, cells were incubated with fluorescent phalloidin (Alexa Fluor 488 phalloidin, Thermo-Fisher Scientific, Waltham, MA, USA) for 1 h at room temperature and then washed with PBS. DNA was stained with DAPI solution (Invitrogen, 1:1000 in PBS) for 10 min and slides were mounted with Fluoromount Acquous Mounting Medium (Sigma-Aldrich). The immunofluorescence (IF) signals of RRAD were acquired for each experimental group with a confocal microscope by Leica (Wetzlar, Germany) (DMI3000B equipped with a Sp5 laser-scanning confocal system) using a 63× oil immersion objective. For evaluating nuclear and cell size changes, images were acquired using an epi-fluorescence microscope with a 20× objective (AXIOVERT 200 Fluorescent, Carl Zeiss, Jena, Germany). Nuclear or cell size measurements and nuclear morphometric analysis were performed with the Image J software v.2.1.0 [25].

### 2.7. RNA Isolation and Retrotranscription

Total RNA was extracted from VSMCs using the Direct-zol^TM^ RNA MiniPrep Plus kit (Zymo Research, Irvine, CA, USA). The concentration and purity of RNA were measured using the Nanodrop 1000 spectrophotometer (Thermo-Fisher Scientific). All RNA samples had an A260/280 value of 1.8–2.1. The quality of RNA was also evaluated using the Tape Station 2200 instrument (Agilent, Santa Clara, CA, USA). All the samples had a RIN value ≥ 9. One μg of total RNA was treated with the RQ1 RNase-Free Dnase (M6101, Promega, Madison, WI, USA) and then cDNA was synthetized in 20 μL reactions using the High-Capacity cDNA Reverse Transcription Kit (4368814, Applied Biosystems, Waltham, MA, USA), according to the manufacturer’s instructions. Alternatively, the RNA samples were retrotranscribed using the iScript gDNA Clear cDNA Synthesis kit (1725035, BIO-RAD, Berkley, CA, USA).

### 2.8. Selection of Genes and lncRNAs Associated with Senescence

A panel of 24 transcripts, including protein-coding genes and lncRNAs, was manually selected from the literature according to their association with senescence and by querying specific online resources such as The Human Ageing Genomic Resources (HAGR) (https://genomics.senescence.info/CellAge (v3, accessed on 23 April 2023)), Reactome (https://reactome.org/ (v.84, accessed on 29 March 2023)) and Human Protein Atlas (HPA) (https://www.proteinatlas.org/ (v.23, accessed on 23 April 2023)). A set of 12 protein–coding genes was selected as involved in different functions such as cell cycle, DNA damage, and SASP, as well as novel biomarkers (Appendix A). For lncRNAs, we focused on a list of 12 lncRNAs (Table 1) involved in several pathways and in the expression and secretion of SASP components.

### 2.9. Quantitative RT-PCR (qRT-PCR)

Quantitative RT-PCR was performed with the QuantStudio 5 thermocycler (Applied Biosystems) in 384-well plates using the GoTaq qPCR Master Mix (A6002, Promega). A total of 10 μL PCR reactions were prepared containing 2 μL of reverse transcriptase product and 0.2 μL of each primer (10 µM) for specific genes (Appendix A). The PCR mixtures were initially denatured at 95 °C for 2 min, followed by 40 cycles of 95 °C for 10 s and 60 °C for 1 min. The results were analyzed with the QuantStudio Design & Analysis Software v1.5.2 (Applied Biosystems). The melting curve showed a single product peak, indicating good product specificity. The calculation of gene expression levels was based on the ΔΔCt method, using the geometric mean of the expression values of three normalizer genes (*CYC1*, *EIF4A2*, and *RPSA*), chosen because they are more stable under the experimental conditions. The stability of the three normalizer genes was evaluated by comparing their expression in two conditions (young and old) and biological replicates of two independent cultures for each condition (Appendix A). Fold changes were calculated using 2-(ΔΔCt) and comparing senescent versus young/proliferating cells.

### 2.10. Protein Isolation, Quantification, SDS Page, and Western Blot

For the preparation of total cell lysates, cells were washed with ice-cold PBS and lysed with lysis buffer (NaCl 150 mM, TRIS 50 mM pH 7.6, NONIDET P-40 0.5%, and protease inhibitors (Merck, Milan, Italy)). Protein concentration was determined using a Pierce BCA Protein Assay Kit (Pierce, Rockford, IL, USA), and 15 μg of samples were run on SDS-PAGE. To analyze the secretion of inflammatory markers, cells were incubated for 72 h with a serum-free medium. Aliquots of media were loaded after correcting for cell protein content and run on SDS-PAGE. The different proteins (PCNA, LMNB1, p21, P53, RRAD, IL1β, IL6, and MMP3) were detected using specific primary and secondary antibodies as needed (Appendix A, which also contains antibody dilutions). Quantification of western blot bands was performed by densitometric analysis using the Image Studio Lite software v 3.1 from Li-Cor Bioscience (Lincoln, NE, USA).

### 2.11. Statistical Analysis and Data Visualization

Data are presented as the mean ± SD of at least 3 experiments, each performed in triplicate and analyzed with GraphPad Prism 9 software. The analysis was performed with the unpaired Student’s *t*-test with Welch correction or 2-way ANOVA, followed by Šidák’s multiple comparisons test. Statistical significance was set at *p* < 0.05. Data on lncRNA expression refer to two different experiments, each performed in triplicate. Statistical analysis was performed with the Mann–Whitney test using the Holm-Šídák method and with *p* < 0.05 as the significance threshold. The actual *p* values of all experiments are reported in Appendix A. The heatmap of fold change (FC) qRT-PCR data was created by the Morpheus online tool (https://software.broadinstitute.org/morpheus/, (accessed on 23 April 2023)).

## 3. Results

Before analyzing the expression of new senescence-associated markers, we needed to establish a replicative senescence cellular model by serially passing human VSMCs [34]. We analyzed two different groups of cells: VSMCs that have been serially passaged 5–7 times (representing the “young” proliferating cells), and VSMCs that have been passaged 15–17 times (representing the “old/senescent” non-proliferating cells). Of note, after being detached and plated for the 15th–17th time, cells did not attach to the cell plate or survive.

### 3.1. Senescent VSMCs Show Increased SA-β-Gal Activity

First, we evaluated the activity of SA-β-gal, which is a lysosomal hydrolytic enzyme up-regulated in senescent cells [35]. As shown in Figure 1A,B, at least 50% of old cells are positive for SA-β-gal staining (indicated by the blue color) compared to about 20% of young cells, indicating increased SA-β-gal activity in senescent VSMCs. We also evaluated the expression of the β-gal encoding gene *GLB1* by qRT-PCR. The *GLB1* gene expression level in old cells is 50% higher compared to young cells (Figure 1C).

### 3.2. Senescent VSMCs Display Altered Cell and Nuclear Morphology

Senescence is normally accompanied by significant morphological changes at the cell as well as nuclear levels [36,37]. As shown in Figure 2A, old/senescent cells become flat, enlarged, and vacuolized compared to young ones.

Nuclear changes have been well documented in senescent cells, which show prominent and, at times, multiple nuclei, with severe chromatin condensation detected as large and fluorescent after staining with DAPI [38]. To evaluate the nuclear area and shape, cells have been stained with DAPI and F-actin. As shown in Figure 2B,C, old cells show a statistically significant increase in cell and nuclear area compared to young cells.

A nuclear morphometric analysis (NMA) was also performed using a FIJI plug-in [13,25]. This approach allowed us to divide cells into various groups based on their nuclear morphometry. These groups included: normal, irregular, small regular (apoptotic), small (mitotic), small irregular, large regular (senescent), and large irregular nuclei [13,25]. We observed a reduced percentage of normal nuclei in old compared to young cells (88% vs. 66%), as well as an increased percentage of large and regular (senescent; 29% vs. 8%) and irregular nuclei (5% vs. 3%) (Figure 2D).

### 3.3. Senescent VSMCs Display Inhibited Proliferation and Cell-Cycle Arrest

One feature of senescent cells is their exhausted replicative potential [7,8,9]. Therefore, we evaluated the expression of the proliferating cell nuclear antigen (PCNA), an aging biomarker that is involved in DNA repair and whose expression is decreased in aged cells [39]. As shown in Figure 3A, old VSMCs have a significantly reduced *PCNA* expression at both mRNA (60% reduction of mRNA levels, *p* < 0.001 vs. young) and protein (65% reduction, *p* < 0.05 vs. young) levels. This led to an inhibition of cell proliferation (Figure 3B), and a much longer doubling time for old cells (67 vs. 27 h in young cells). As expected, when we evaluated the cell cycle, we observed an accumulation of old VSMCs in the G1 phase (Figure 3C) [9,40]. In addition, old VSMCs have dramatically reduced migratory activity (70% reduction, *p* < 0.01 vs. young cells) (Figure 3D).

### 3.4. Senescent VSMCs Express Well-Known Senescence Biomarkers

Next, we evaluated in young and old cells the expression of lamin B1 (LMNB1), a protein component on nuclear lamina whose downregulation causes a detachment of chromatin domains normally attached to the nuclear lamina, leading to the redistribution of hetero-chromatin from the nuclear periphery into the interior. This may trigger senescence, as reported before [41,42]. Figure 4A demonstrates that old VSMCs have a significantly reduced LMNB1 expression at both mRNA (65% reduction, *p* < 0.01 vs. young) and protein (60% reduction, *p* < 0.05 vs. young) levels, indicating an altered nuclear membrane in old cells.

During senescence, SASP-related chromatin folding, and RNA homeostasis are coordinated by the extracellular senescence factor high-mobility group box 1 (HMGB1) [43]. As shown in Figure 4B, *HMGB1* mRNA levels are significantly reduced in old VSMCs. We also observed an upregulation of the expression of p53, whose activation regulates the expression of a large set of genes involved in cell-cycle arrest [44]. We observed a doubling of the expression of the p53 protein, although the mRNA levels were not affected (Figure 4C,D). In line with this data, old senescent cells showed an increased expression of p16 and p21, two tumor suppressors that, by preventing retinoblastoma phosphorylation, regulate the cell cycle, leading to cell growth arrest (Figure 4C,D).

Having established an RS model in human VSMCs that replicates several classical senescence-associated markers, we pursued new possible markers of aging VSMCs. To this end, we included in our list of protein-coding genes (Appendix A) the ras-related glycolysis inhibitor and calcium channel regulator (RRAD) which has been recently shown to be a negative regulator of cellular senescence [45]. In our model, we observed a consistent and statistically significant upregulation of *RRAD* mRNA levels in senescent VSMCs (fold change > 6 in old cells vs. young cells; Figure 5A), although the analysis at the protein level did not show a similar direction (Figure 5B). As reported in the Human Protein database (HPA), the screening of *RRAD* gene expression levels in 44 human normal tissues (tissue atlas), as well as in 15 different cell types (single cell atlas), indicates an RRAD-specific expression in tissues and cells belonging to the musculoskeletal and cardiovascular systems (https://www.proteinatlas.org/ENSG00000166592-RRAD, accessed on 24 April 2023). Although *RRAD* expression was reported in SMCs in the HPA database, its subcellular localization is not publicly available. Using confocal microscopy, we showed RRAD localization both in the cytoplasm and in the nucleoplasm (Appendix A), both in young cells and in old/senescent VSMC cells. Of note, the signal of RRAD appeared more dispersed and diffuse in the old cells, possibly due to their altered cellular morphology.

### 3.5. Senescent VSMCs Exhibit a Change in Inflammatory Biomarkers

Senescent cells exhibit significant changes in their secretome including the expression of a variety of inflammatory proteins such as cytokines and interleukins [46], a phenomenon known as SASP [47]. To complete the characterization of our RS model, we evaluated the gene expression of different inflammatory markers that are typical of the SASP. The mRNA levels of *IL1β*, *IL6*, *IL8*, and *MMP3* are significantly increased up to nine-fold in old cells (Figure 6A). A significant increase in IL1β, IL6, and MMP3 protein levels is also displayed by senescent VSMCs (Figure 6B). The increased secretion of MMP3 was confirmed in conditioned media of senescent VSMCs (Figure 6C).

### 3.6. LncRNAs PURPL and NEAT1 Are Overexpressed by Senescent VMSCs

To further expand the characterization of senescent VSMCs, we evaluated the expression of a panel of lncRNAs. LncRNAs have recently drawn attention as important regulators of pathways associated with senescence also displaying altered expression levels in various models of cellular senescence [20]. Based on the literature, we selected 12 lncRNAs involved in senescence (see Table 1 and Section 2) and we measured their expression levels using qRT-PCR in old and young VMSCs. Our quantitative analysis revealed a significant strong up-regulation of two lncRNAs, namely P53-upregulated regulator of P53 levels (*PURPL*; fold changes ranging from 3.2 to 22.5, mean average = 15.21) and nuclear paraspeckle assembly transcript 1 (*NEAT1*; FC ranging from 1.7 to 7.9, mean average = 4.18) (Figure 7). For the remaining lncRNAs, we observed a more heterogeneous behavior in senescent VSMCs due also to their low expression levels.

## 4. Discussion

Our current study provides new insights into the different processes that regulate aging in human aortic VSMCs. In the search for new senescence-associated markers, we first created a cellular model of replicative senescence by serially passing human VSMCs. In our experimental conditions, the old/senescent VSMCs express many of the typical senescence-associated markers already described in the literature. In this cellular model of VSMC senescence, we demonstrated the enhanced expression of RRAD and the lncRNAs PURPL and NEAT1.

Old VSMCs display many of the classical senescence markers: an increased percentage of SA-β-gal expressing cells and a different morphology, showing a flat and enlarged cell body and the presence of vacuolization, a reduced percentage of normal nuclei, and a parallel increase in irregular, enlarged, and senescent nuclei characterized by chromatin condensation. Old cells have an exhausted proliferation potential, displaying a much longer doubling time, and accumulate in the G1 phase of the cell cycle. Moreover, the expression of the aging biomarker PCNA, which is involved in cell proliferation and DNA repair, is decreased by up to 65% at both mRNA and protein levels. We also observed a significant reduction in the expression of LMNB1. We observed an increased expression of p53, a transcription factor that can both upregulate or downregulate the expression of specific target genes by binding their promoter region [44]. In addition, the expression of cell-cycle inhibitors p16 and p21 is stimulated by more than 200% and 400%, respectively, compared to young VSMCs.

The high heterogeneity of senescent cells has been demonstrated by many papers [13,48,49]. Moreover, senescence has been described as a multistep process that could be divided into four phases (quiescence or temporary senescence, early senescence or stable growth arrest, full senescence (SASP expression) and late senescence (phenotypic diversification) [50]. Our data could reflect the presence of a mixture of cells in our experimental model. Part of the cells could be fully senescent, with a stable growth arrest, while a few others could still be in the early phases of quiescence.

A general agreement about senescent cells is the significant changes in their secretome, including the expression of a variety of molecules such as inflammatory proteins, growth factors, degradative enzymes like MMPs, and insoluble proteins or extracellular matrix components [46]. Moreover, during senescence, SASP-related chromatin folding and RNA homeostasis are coordinated by the extracellular senescence factor HMGB1 [43], an important chromatin protein that is secreted by stressed cells and serves as an alarmin, cytokine, or growth factor to activate the immune response. Suppression of HMGB1 induces cell-cycle arrest and senescence in association with p21 upregulation [51]. Following these observations, in our old VSMCs, HMGB1 expression is significantly reduced by 40%, and in parallel, we observed strikingly increased levels of inflammatory markers, such as IL1β, IL6, IL8, and MMP3, at both mRNA and protein levels. The changes in the secretome of old VSMCs are a typical feature of the late senescent cellular state. Taking into account all the cellular and molecular biomarkers, we consider that VSMCs at passages 15–17 (the old cells in our paper) acquired a senescent phenotype.

In our cell model, we observed a significant upregulation of *RRAD* mRNA levels. To our knowledge, this is the first report showing an increased expression of *RRAD* in senescent human VSMCs. RRAD is predicted to be involved in small GTPase-mediated signal transduction, it has been implicated in some types of cancer [52] and is considered a biomarker of congestive heart failure [53,54]. *RRAD* expression appears to be stimulated by oxidative stress [55], and it has been recently associated with cellular senescence in human skin fibroblasts as a negative regulator [45]. It has been proposed that increased levels of RRAD may serve as a negative feedback mechanism in the effort to reduce the level of ROS, thus countering cellular senescence [45]. In our VSMC replicative senescence model, *RRAD* gene expression level was upregulated in old cells, although western blot analysis showed lower protein expression. The correlation between mRNA and protein expression is affected by multiple layers of regulation and, therefore, it is not surprising to observe contrasting results [56]. We hypothesized that the increased gene expression of *RRAD* in old VMSCs is the response to the oxidative stress induced by cellular senescence. To determine if the fate of the RRAD protein in old cells is altered, the study of protein stability would be needed. Intriguingly, the investigation of the subcellular localization of RRAD in young and senescent VSMC cells using confocal microscopy indicates a more dispersed and diffuse distribution in old cells. A deep biochemical analysis of RRAD in our model of senescence will be the objective of further investigations.

Among the set of 12 lncRNAs manually selected for being involved in senescence, we detected with qRT-PCR a significant up-regulation of *PURPL* and *NEAT1* in old cells compared to the young ones. *PURPL* is known to be a p53 target, as its promoter contains p53-response elements. In colorectal cancer cells, *PURPL* is transcriptionally activated by p53 and, in return, it can decrease the levels of p53 and its targets, such as p21. Mechanistically, PURPL can negatively regulate p53 stability by inhibiting its interaction with the MYBBP1A protein, which can bind and stabilize p53 [30]. This negative regulation of p53 by PURPL was also observed in melanoma cells, where PURPL was shown to repress autophagic cell death by associating with mTOR and modulating ULK1 phosphorylation [57]. On the contrary, in liver cancer cells, *PURPL* expression is still activated by p53, but its negative effect on p53 levels is lost. Instead, upon p53 activation, loss of PURPL leads to downregulation of genes involved in mitosis, indicating a role of PURPL in cell-cycle progression and mitosis [58]. Up-regulation of *PURPL* was consistently observed in several cellular models of replicative and induced senescence, including a model of ionizing-radiation-induced senescence of human aortic endothelial cells, which is the other major cell type present in the aortic wall [31].

We also found the lncRNA *NEAT1* to be significantly upregulated in senescent VMSCs. *NEAT1* is also a p53 target and it is a core structural component of paraspeckles, nuclear bodies with a key role in gene expression regulation through several mechanisms, including regulation of transcription, regulation of translation, and modulation of miRNA processing [59,60,61,62]. NEAT1 has been also associated with diseases such as cancer, immune inflammation, and neurodegeneration [63]. Several lines of evidence show that *NEAT1* expression, as well as paraspeckle formation, is induced by p53 and that NEAT1 is part of an autoregulatory negative feedback loop that attenuates p53 activity [64,65,66]. NEAT1 is also able to inhibit p21 expression by guiding the epigenetic repressor enhancer of zeste homolog 2 (EZH2) to the p21 promoter [67]. On the other hand, it was shown to activate pro-inflammatory cytokine IL8 transcription [61]. Importantly, NEAT1 has a key role in VSMCs switching from a contractile to a proliferative phenotype by epigenetically repressing the expression of smooth muscle-specific genes [68]. In our work, increased levels of p53 and p21 were observed in senescent VMSCs, as expected in a state of cell-cycle arrest. In this context, we hypothesize that *PURPL* and *NEAT1* up-regulation may be part of a negative feedback response to buffer excessive p53 and downstream target levels and activity, and partly try to restore proliferative capacity. At the same time, NEAT1 may promote the SASP by increasing IL8 levels. Overall, the understanding of the role of lncRNAs in the cardiovascular context and aging is emerging, so further investigations must be implemented.

Senescence may exert both beneficial and negative effects after tissue injury, depending on the context. Although senescent cells may maintain healthy physiology, senescence of various cell types has been implicated in the pathogenesis of atherosclerosis, including endothelial cells, VSMCs, macrophages, and T cells. The accumulation of senescent VSMCs contributes to aging, as well as age-related diseases of the cardiovascular system [34]. VSMCs were reported to be one of the key pro-inflammatory senescent cell populations and were found in unstable, rupture-prone atherosclerotic plaques [69,70]. Like other aging cells, senescent VSMCs have a low ability of mitotic division and show changes in cell signaling pathways and senescent markers, such as SA-β-gal activity, levels of p16, p38, p53, p21, and express the SASP [10]. Due to the loss of proliferative and migration potential of senescent cells, senescence can trigger plaque instability directly by reducing the VSMC content of the fibrous cap and compromising its repair after rupture [5]. However, more recent evidence suggests a more active role of VSMC senescence in promoting plaque destabilization by driving plaque inflammation, matrix degradation, and defective autophagy [5,71]. In this scenario, the involvement of lncRNAs in atherosclerosis is emerging, by regulating several key processes such as cholesterol homeostasis, vascular inflammation, VSMC phenotypic switch, and cell death, among others [68,72,73]. As discussed above, upregulation of *NEAT1* in senescent VSMCs may be a fundamental contributor to plaque inflammation and destabilization by setting a ‘macrophage-like’ state with increased SASP levels. Interestingly, *NEAT1* is involved in postischemia myocardial remodeling [74].

## 5. Conclusions

Cell senescence could be targeted to prevent or delay the aging process and tissue dysfunction extending lifespan [75,76]. Some biochemical agents with anti-senescence potential called ‘senolytics’, compounds that selectively target and eliminate senescent cells by inducing apoptosis, have been tested in the prevention/treatment of diseases [1]. On the other hand, an alternative approach to target senescent cells would be the use of senomorphics that inhibit the detrimental effects of SASP secreted by senescent cells without killing them [77]. However, we need better senescence-associated targets, which may also include lncRNAs, to ameliorate our therapeutic approach. In our experimental settings, we characterized the expression of new promising cell-specific molecular markers in senescent human VSMCs. It is basic knowledge that cellular senescence is the result of complex interplay of various biological and metabolic cell- and tissue-specific processes, so our experimental in vitro characterization is still incomplete. An in vivo model of senescence would be more than useful, but it is outside the scope of the present manuscript. The definition of the functional role of PURPL, NEAT1, and RRAD in senescent human VSMCs, is also beyond the scope of the present study but future investigations are planned. However, we think that these newly discovered markers modulated in senescent VSMCs, although in an in vitro model, are promising molecular tools. They could be used in combination with landmark senescence markers to better discriminate senescent VSMCs implicated in atherogenesis and atherosclerosis complications and may help in designing new strategies for promoting healthy vascular aging.

## Figures and Tables

**Figure 1 biomedicines-11-03228-f001:**
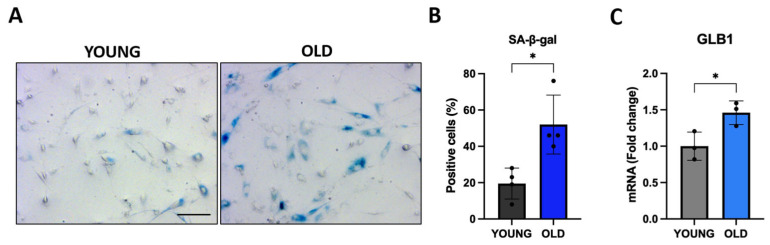
Senescence-associated β-gal activity and *GLB1* expression in young and old human VSMCs. Cells have been plated and the activity of SA-β-gal and *GLB1* expression have been evaluated. (**A**) Representative images of SA-β-gal staining in young and senescent VSMCs (scale bar 100 μm). The histograms indicate the percentage of (**B**) SA-β-gal positive cells (2000 young and 1700 old cells were analyzed) and (**C**) *GLB1* mRNA levels in young (black or grey) and old (blue or light blue) cells. Data were compared with an unpaired *t*-test with Welch’s correction: * *p* < 0.05 vs. YOUNG.

**Figure 2 biomedicines-11-03228-f002:**
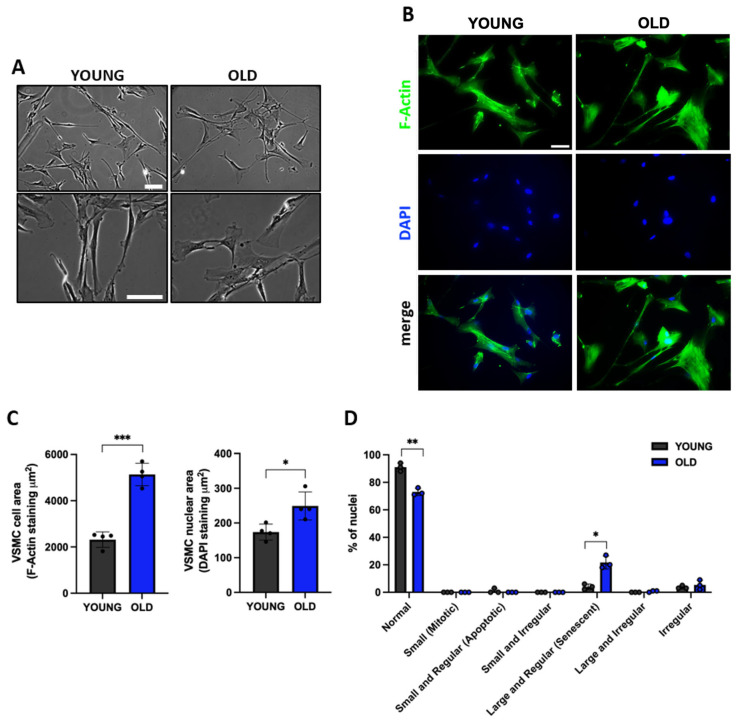
Morphological and nuclear changes in old compared to young VSMCs. (**A**) Representative pictures of cell morphology, at two different objective lens magnifications (10× and 20×) and (**B**) F-Actin staining (scale bar 50 μm) visualized with an epi-fluorescence microscope. (**C**) The graphs show the mean ± SD of cell and nuclear area, analyzed in 4 independent experiments (800 young and 500 old cells were analyzed). (**D**) Nuclear morphometric analysis of young and old VSMCs. Unpaired *t*-test with Welch’s correction. * *p* < 0.05, ** *p* < 0.01, *** *p* < 0.001 vs. YOUNG.

**Figure 3 biomedicines-11-03228-f003:**
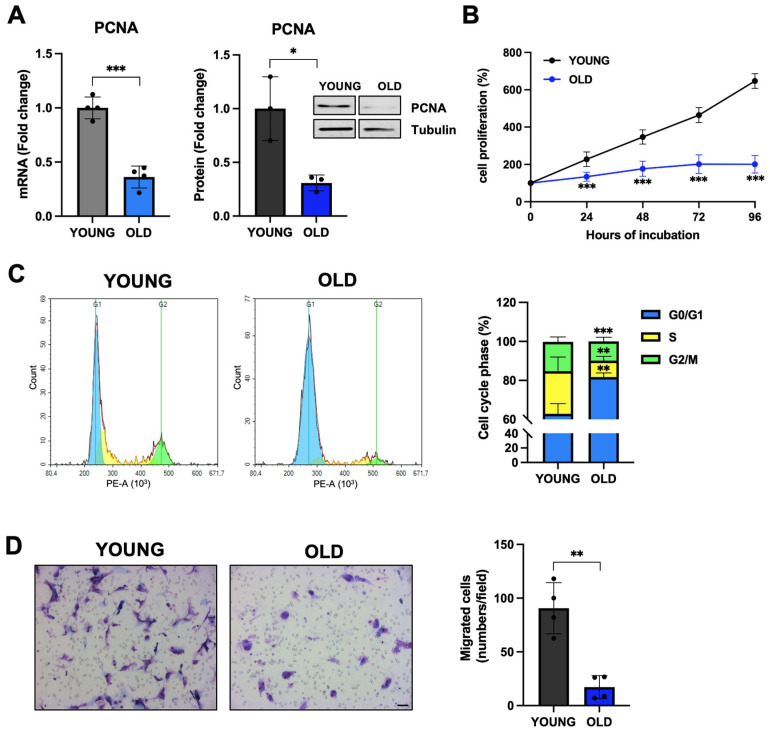
Proliferative state of young and old VSMCs. (**A**) Expression levels of the proliferative marker PCNA have been measured using q-RT-PCR and western blot. Young (grey for mRNA or black for protein) and old (light blue for mRNA or blue for protein). Data were analyzed with an unpaired *t*-test with Welch’s correction. (**B**) The graph depicts the curve of cell proliferation for young and old cells. Data were analyzed using 2-way ANOVA, followed by a Sidak post hoc test. (**C**) The histograms show the percentage of old and young cells in the different stages of the cell cycle. Data were analyzed with an unpaired *t*-test with Welch’s correction. (**D**) Cell migration was assessed using the Boyden chamber and the number of migrated cells was manually counted by staining with a Diff-Quik staining set (scale bar 50 μm). Results were analyzed with an unpaired *t*-test with Welch’s correction. * *p* < 0.05; ** *p* < 0.01; *** *p* < 0.001 vs. YOUNG.

**Figure 4 biomedicines-11-03228-f004:**
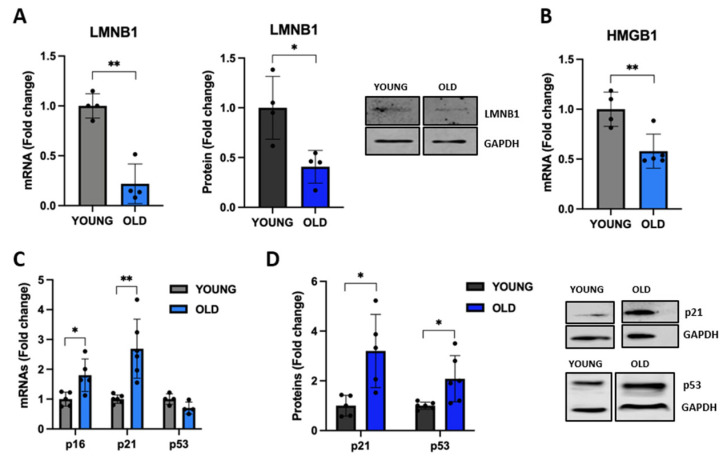
Expression of known senescence-associated biomarkers. The expression levels of LMNB1 (**A**), HMGB1 (**B**), and the cell-cycle inhibitors p16, p21, and p53 (**C**,**D**) have been analyzed in young and old VSMCs. Young (grey for mRNA or black for protein) and old (light blue for mRNA or blue for protein). The histograms report the mean ± SD of data. For western blot experiments, the representative bands are shown with the corresponding housekeeping protein. Unpaired *t*-test with Welch’s correction: * *p* < 0.05; ** *p* < 0.01 vs. YOUNG.

**Figure 5 biomedicines-11-03228-f005:**
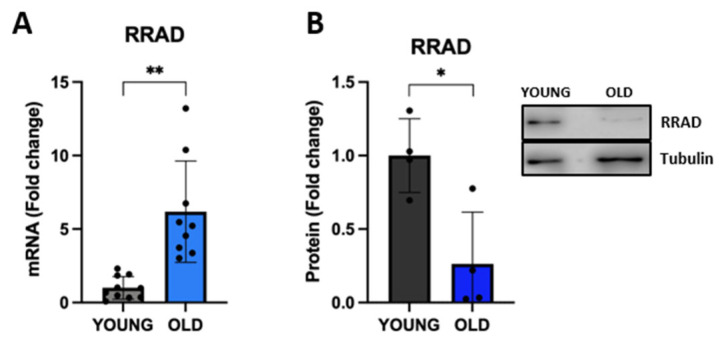
Expression of the new senescence-associated biomarker RRAD. The expression of RRAD was analyzed using qRT-PCR (**A**) and western blot (**B**). Young (grey for mRNA or black for protein) and old (light blue for mRNA or blue for protein). Representative blots are reported in the caption. The histograms report the mean ± SD of data. The representative bands of RRAD are shown with the corresponding housekeeping protein. Unpaired *t*-test with Welch’s correction: * *p* < 0.05; ** *p* < 0.01 vs. YOUNG.

**Figure 6 biomedicines-11-03228-f006:**
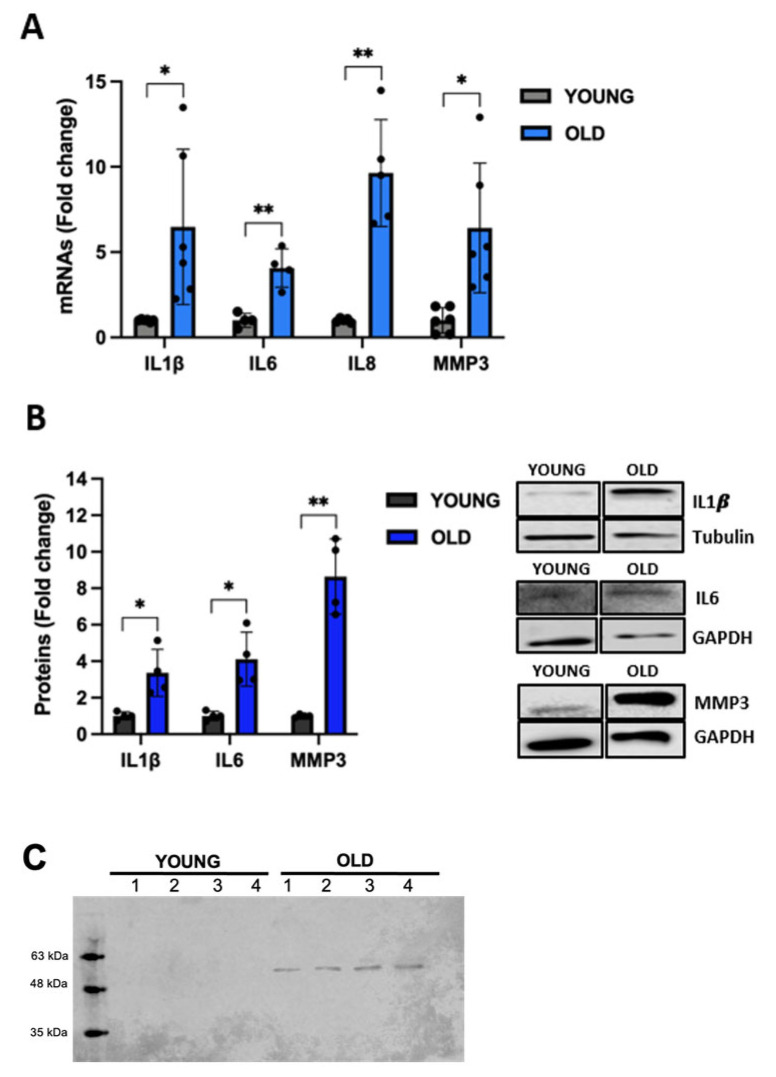
Expression of SASP markers in young and old VSMCs analyzed at both mRNA (**A**) and protein (**B**) levels using qRT-PCR and western blot, respectively. The secretion of MMP3 was evaluated using a western blot in conditioned media from young and old VSMCs (**C**). Unpaired *t*-test with Welch’s correction: * *p* < 0.05; ** *p* < 0.01 vs. YOUNG.

**Figure 7 biomedicines-11-03228-f007:**
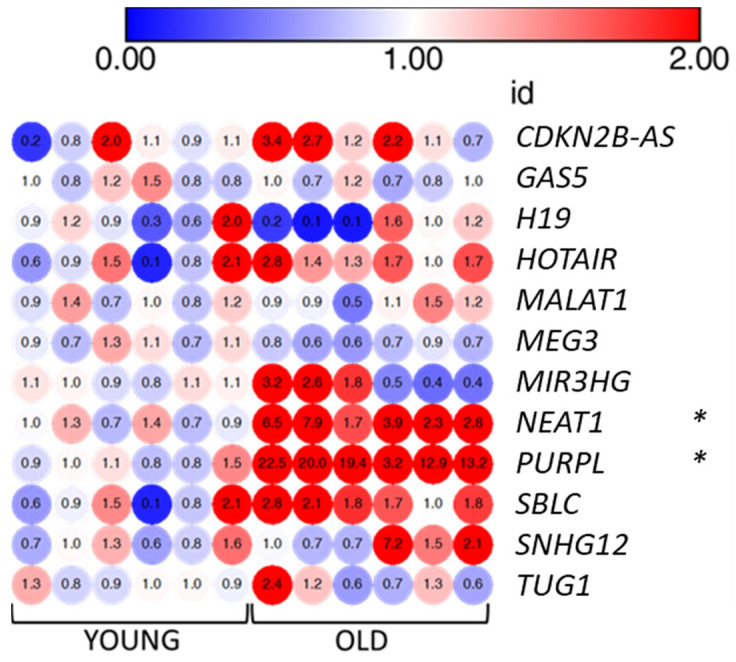
Heatmap of gene expression levels of selected 12 lncRNAs. The fold change values of young cells and old cells are indicated in each dot of the heatmap (6 replicates for condition). Fold changes were calculated using 2^−(ΔΔCt)^ and comparing senescent versus young cells. Blue and red color grade dots indicate levels of down-regulation and upregulation. Mann–Whitney test: * *p* < 0.05; vs. YOUNG. The heatmap was created with the Morpheus online tool (https://software.broadinstitute.org/morpheus/, accessed on 23 April 2023).

**Table 1 biomedicines-11-03228-t001:** List of lncRNAs selected for this study and their published roles in senescence.

lncRNA Symbol	lncRNA Name	Expression in Cellular Senescence	Function/Mechanism of Action	References
*CDKN2B-AS1*	CDKN2B antisense RNA 1	Up/Down	Suppresses the expression of CDKN2A/p16 and CDKN2B/p15 by recruiting the repressive Polycomb complexes.	[20]
*H19*	H19 imprinted maternally expressed transcript	Up/Down	Inhibits p53, CDKN1C, and IL6/STAT3/p21 pathways (anti-senescence function). Derepresses β-catenin (pro-senescence function).	[20]
*HOTAIR*	HOX transcript antisense RNA	Up	Activates NF-κB/IL6 and p53/p21 pathways through DNA damage response.	[20]
*GAS5*	Growth arrest-specific 5	-	Sponges miR-223, which inhibits the anti-senescence NAMPT enzyme.	[22]
*MALAT1*	Metastasis-associated lung adenocarcinoma transcript 1	Down	Controls cell-cycle progression by regulating p53 activity and the expression of B-MYB transcription factor.	[26]
*MEG3*	Maternally expressed 3	Up	Enhances p53 transcription and reduces p53 degradation. Promotes p53 binding to target promoters.	[20,27]
*MIR31HG*	MIR31 host gene	Up	Activates the expression and secretion of SASP components.	[28]
*NEAT1*	Nuclear paraspeckle assembly transcript 1	-	Facilitates the expression of IL8.	[29]
*PURPL*	p53 upregulated regulator of p53 levels	Up	Negatively controls p53 levels by interfering with the p53-MYBBP1A complex.	[30,31]
*SBLC*	Senebloc	Down	Promotes p53 degradation. Mediates epigenetic silencing of p21 through regulation HDAC5.	[32]
*SNHG12*	Small nucleolar RNA host gene 12	-	Inhibits p16, p21, and γH2AX expression. Regulates DNA damage response via interaction with DNA-PK kinase.	[33]
*TUG1*	Taurine up-regulated 1	Up	Growth arrestor induced by p53 upon DNA damage. Inhibits the pro-proliferation HOXB7 transcription factor	[29]

## Data Availability

The data presented in this study are available on request.

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
