# Peer review of "PURPL and NEAT1 Long Non-Coding RNAs Are Modulated in Vascular Smooth Muscle Cell Replicative Senescence"

_biomedicines, 2023, doi:10.3390/biomedicines11123228_

Round 1

Reviewer 1 Report (Previous Reviewer 1)

Comments and Suggestions for Authors

Using cultured human aortic smooth muscle cells obtained from ATCC and cultured for 5-7 (young cells) and 15-17 (senescent cells) passages, Rossi et al. studied how the old culture is different from the young culture in order to characterize senescent vascular smooth muscle cells (VSMCs).  The senescent cells exhibited various characteristics expected in old cultures.  The morphological, biochemical, and molecular biological characteristics chosen for the study were those already known to be markers of senescent cells.  For this reason, this is not a study for mining new markers.  This is a disappointing aspect of the study and is a weakness.  Although the authors state that senescence in VSMCs has not been studied, this is a rather minor novelty of the study, and for this reason, the significance of the study is limited.  Some specific comments are listed below.

1.     Fig. 2B.  These images cannot be true confocal images.  Cultured SMCs are known to express rich stress fibers, especially cells in old cultures.  Please consult a confocal imaging tech/expert and provide high quality confocal micrographs.

2.     Page 10, line 2.  “In fact, we observed a doubling of the expression of p53 protein, although the mRNA levels were slightly reduced (Figure 4C, D).”  The data analysis of the “slight reduction” in the mRNA level referred to here is statistically insignificant.  As such, one must conclude that there is no difference (against the temptation of seeing difference!).

3.     In several places, Supplemental materials are cited, but they are not included in the reviewer packet.  If the confocal images are as poor as those in Fig. 2B, please provide high quality micrographs.

4.     SASP by definition refers to materials that are secreted, but the data shown in Fig. 6B appear to come from the whole cell.  Please show increased  levels of these molecules in the media.

Comments on the Quality of English Language

Careful editing is requested.  Several punctuation errors were noted.

Author Response

Reviewer 1

We thank the reviewer for the remarks and we apologize for not being able to explain more clearly what was the aim of our manuscript. We hope we succeeded this time. We have revised the entire paper, taking into consideration all the Reviewer’s comments. More in detail:

Reviewer 1’s comments and suggestions for authors

 Using cultured human aortic smooth muscle cells obtained from ATCC and cultured for 5-7 (young cells) and 15-17 (senescent cells) passages, Rossi et al. studied how the old culture is different from the young culture in order to characterize senescent vascular smooth muscle cells (VSMCs).  The senescent cells exhibited various characteristics expected in old cultures.  The morphological, biochemical, and molecular biological characteristics chosen for the study were those already known to be markers of senescent cells.  For this reason, this is not a study for mining new markers.  This is a disappointing aspect of the study and is a weakness.  Although the authors state that senescence in VSMCs has not been studied, this is a rather minor novelty of the study, and for this reason, the significance of the study is limited. 

 We apologize to the reviewer since we were not able to clearly explain what was the actual aim of our study. We did not try to replicate the data already present in the literature showing the expression of several already well-known senescence-associated markers in our replicative senescence model. Since there are no unanimously agreed senescence markers in human VSMCs, to improve our knowledge we looked for new possible senescence markers. However, for reaching our goal we needed to start from a well-characterized cellular model of replicative senescence. Therefore, we first carefully characterized the young/not-senescent versus the old/senescent VSMCs. Next, in the very same models we evaluated the expression of novel senescence markers.

 Some specific comments are listed below.

  1. 2B.  These images cannot be true confocal images.  Cultured SMCs are known to express rich stress fibers, especially cells in old cultures.  Please consult a confocal imaging tech/expert and provide high quality confocal micrographs.

We did not describe with sufficient clarity the experiments shown in Fig 2B and there has been a misunderstanding. The images shown in Figure 2B were not taken with a confocal microscope, but they were taken with an epifluorescence microscope. The Figure legend has been modified accordingly.

  1. Page 10, line 2.  “In fact, we observed a doubling of the expression of p53 protein, although the mRNA levels were slightly reduced (Figure 4C, D).”  The data analysis of the “slight reduction” in the mRNA level referred to here is statistically insignificant.  As such, one must conclude that there is no difference (against the temptation of seeing difference!).

We agree with the reviewer and modified the sentence accordingly.

  1. In several places, Supplemental materials are cited, but they are not included in the reviewer packet.  If the confocal images are as poor as those in Fig. 2B, please provide high quality micrographs.

We apologize for this unfortunate matter, but we took good care in preparing a file containing all the supplementary materials (5 tables and 1 suppl. Figure) and we did attach the file during the resubmission process. However, due to unknown reasons, this file was not attached to the reviewer's package. We do not have an explanation for this. We contacted the editor and she could not give us an answer either. We are attaching again the Supplementary material file and we hope that this time everything will go smoothly.

  1. SASP by definition refers to materials that are secreted, but the data shown in Fig. 6B appear to come from the whole cell.  Please show increased levels of these molecules in the media.

We agree with the reviewer’s comment. Therefore, we performed new experiments and analyzed by western blot the expression of inflammatory markers in conditioned media. As shown in the new Figure 6 panel C, old/senescent VSMCs did show an increased expression of MMP3, while young/non-senescent VSMCs did not express any MMP3. We also tested the secretion of IL1b and IL6, but we did not see any bands. This could be due to the very low amount of secreted interleukins or to the sensitivity of the antibodies we used.

Comments on the Quality of English Language

Careful editing is requested.  Several punctuation errors were noted.

We completely revised the text and corrected the mistakes.

Reviewer 2 Report (Previous Reviewer 3)

Comments and Suggestions for Authors

In this study, the authors characterized a cellular model of replicative senescence in human VSMCs by means of multi-biomarkers approaches, evaluating the expression of manually selected senescence-associated genes and lncRNAs. The authors concluded that the detection of novel molecular markers of senescence, such as RRAD, PURPL, and NEAT1, could be helpful for future studies on potential anti-aging factors.

Comments

The reviewer has some concerns as follows:

1. The authors did still not respond the reviewer’s previous comments: One of the major concerns is that the images of the western blot analysis are not faithfully presented; compared with the original film provided, the presented images are obviously altered and edited, for example, in Figures 3, 4, and 6, there are 3 protein spots in original gel band, but there are only two protein spots in the presented images in which the middle one is cut off. This transformation method will be considered as a fraudulent method and is not allowed. This needs a major revision. Moreover, please provide enough image data that can be statistically analyzed. 

2. In this study, the presented results cannot support the conclusions.

Author Response

Reviewer 2

Comments and Suggestions for Authors

In this study, the authors characterized a cellular model of replicative senescence in human VSMCs by means of multi-biomarkers approaches, evaluating the expression of manually selected senescence-associated genes and lncRNAs. The authors concluded that the detection of novel molecular markers of senescence, such as RRAD, PURPL, and NEAT1, could be helpful for future studies on potential anti-aging factors.

Comments

The reviewer has some concerns as follows:

  1. The authors did still not respond the reviewer’s previous comments: One of the major concerns is that the images of the western blot analysis are not faithfully presented; compared with the original film provided, the presented images are obviously altered and edited, for example, in Figures 3, 4, and 6, there are 3 protein spots in original gel band, but there are only two protein spots in the presented images in which the middle one is cut off. This transformation method will be considered as a fraudulent method and is not allowed. This needs a major revision. Moreover, please provide enough image data that can be statistically analyzed.

When we resubmitted our revised paper, we provided a clear explanation of how our western blots were designed. The reviewer did not pay any attention to this and we do not think that this is fair behavior. Moreover, no constructive comments or suggestions were given, but criticisms. A reviewer should act to help the authors in improving the quality of their paper.

  1. In this study, the presented results cannot support the conclusions.

For the second time, the reviewer did not say anything about the extensive data we presented but commented only on the quality of our western blot images, suggesting that we have manipulated the data and that this is a fraudulent act. We are quite surprised by the reviewer’s continuous and unacceptable considerations, and we do not accept such types of comments or behaviors again. We provided plenty of data and the reviewer should analyze the data presented, not give insulting comments.

Round 2

Reviewer 1 Report (Previous Reviewer 1)

Comments and Suggestions for Authors

In general, the authors responded well to my comments, except for the fluorescence micrograph in Fig. 2B.  Even as an epi-fluorescence micrographs, the quanity is poor.  Especially in the green images, I cannot see where the focus is.  Everything in these images are out of focus.  Please provide in-focus images.  The fluorescence images in the supplement are of fine, acceptable quality. 

Comments on the Quality of English Language

Much improved.

Author Response

We thank the reviewer for the positive remarks. We changed Figure 2B and we hope that its quality is now acceptable.

Reviewer 2 Report (Previous Reviewer 3)

Comments and Suggestions for Authors

This revised manuscript has a great improvement and can be accepted.

Author Response

We thank the reviewer for the positive comment.

This manuscript is a resubmission of an earlier submission. The following is a list of the peer review reports and author responses from that submission.

Round 1

Reviewer 1 Report

Comments and Suggestions for Authors

Rossi and her colleagues characterized human aortic smooth muscle cells (ASMCs) that were sub-cultured 5-7 times (young ASMCs) and 15-17 times.  The latter culture was thought of as non-proliferating cells.  The study found numbers of molecular and cell biological differences between the two cultures.  A large portion of the data are confirmatory in nature, but they are used to argue that the ASMC culture system represent senescent and non-senescent cell populations.  There are some conceptual issues and confusions regarding what senescent cells are.  The authors state, “Cellular senescence is an irreversible loss of proliferation potential (line 51)”.  There are at least two problems with this statement: 1) it is now known that cell senescence is not irreversible, and 2) Fig. 3B clearly shows these senescent cultures’ ability to proliferate.  The authors must clearly define what cell senescence is and logically justify that the “senescent” ASMC culture used qualifies to be a senescent vascular smooth muscle cell model.  The data presented in this manuscript does not support the title given to this study.  If this ASMC culture reaches the Hayflick limit, then those cells would be senescent.  Listed below are other specific suggestions and comments.

1.       Some methods were not sufficiently described.  Cell culture:  How were cells passaged and plated?  Describe in detail the level of confluency at the time of cell passage, and the number of cells (or the split ratio) plated. Is the frequency of passage determined by the level of confluency or by time (ex. every X days)?  Cell migration:  Describe in more detail the Boyden chamber assay including number of cells plated on the upper chamber, the type of culture medium used in the lower compartment, and how cells were stained.  Also, please see the comment #5.  RNA quantification:  To normalize RNA quantity, CYC1, EIF4A2 and RPSA were used.  Please show that the expression levels of these genes were the same between the old and the young cultures.  Show the data in Fig. 7.

2.       For all the statistical analyses, please show the actual p values, rather than stating p <0.05, etc. Also, where appropriate, state how many cells in total were analyzed for statistical analyses.

3.       Line 204:  (5x and 10 x) must mean objective lens magnifications, not the total magnifications.

4.       Fig. 1A and Fig. 2B:  Out of focus, poor quality images.  Must be replaced by in-focus micrographs.

5.       Fig. 3D.  If the old cells are larger in sizer, wouldn’t this make it harder to go through the membrane?  In general, it is better to trace cell migration directly using time lapse.

6.       Lines 313-314:  The authors state that “a slight down-regulation of MEG3 and up-regulation of SBLC, although not statistically significant, in senescent cells (Figure 7).”  If statistics says no difference, one must conclude no difference.  This statement violates the foundation of statistics.

7.       Fig. S1:  It is possible that RRAD localization in the nucleus is an imaging artifact of superimposition.  Confocal microscopy must be used to unequivocally show intranuclear localization of staining.

Comments on the Quality of English Language

In general fine.  Some typos

Reviewer 2 Report

Comments and Suggestions for Authors

There are several deficits needing to be experimentally addressed to increase the scientific impact of this study.

 1. The study is more descriptive without molecular mechanistic insights.

2. The causal relationship between PURPL and NEAT1 long non-coding RNAs and VSMC senescence as well as the physiological functions of VSMC should be experimentally addressed. Genetic manipulation strategy is highly recommended.

3. In vivo models for VSMC senescence are highly recommended.

4. The image quality of figure 2B was not good, please use the high quality of images.

5. The sample size was too small to use 2-way ANOVA for statistical analysis. A biostatistician is highly recommended for this study.

Comments on the Quality of English Language

There are many typos and grammatic errors in the current form of the manuscript. An English editor s highly recommended.

Reviewer 3 Report

Comments and Suggestions for Authors

In this study, the authors characterized a cellular model of replicative senescence in human VSMCs by means of multi-biomarkers approaches, evaluating the expression of manually selected senescence-associated genes and lncRNAs. The authors concluded that the detection of novel molecular markers of senescence, such as RRAD, PURPL, and NEAT1, could be helpful for future studies on potential anti-aging factors.

Comments

The reviewer has some concerns as follows:

1. One of the major concerns is that the images of the western blot analysis are not faithfully presented; compared with the original film provided, the presented images are obviously altered and edited, for example, in Figures 3, 4, and 6, there are 3 protein spots in original gel band, but there are only two protein spots in the presented images in which the middle one is cut off. This transformation method will be considered as a fraudulent method and is not allowed. This needs a major revision. Moreover, please provide enough image data that can be statistically analyzed. 

2. The young VSMCs were used at the 5-7th passage. Please describe the original passage of VSMCs obtained from ATCC.

3. In general, the presented results cannot support the conclusions. This manuscript needs a major revision for data presentation especially protein expression from Western blot analysis.